

# Geophysical downhole logging analysis within the shallow depth ICDP STAR drilling project (Central Italy)

Paola Montone[1] *, Simona Pierdominici[2], M. Teresa Mariucci[1], Francesco Mirabella[3], Marco Urbani[3], Assel Akimbekova[3], Lauro Chiaraluce[1], Wade Johnson[4] and Massimiliano Rinaldo Barchi[3]

[1]Istituto Nazionale di Geofisica e Vulcanologia, Roma, 00143, Italy
[2]GFZ, German Research Centre for Geosciences, Telegrafenberg, 14473, Potsdam, Germany
[3]Dipartimento di Fisica e Geologia, Università degli Studi di Perugia, Perugia, 06123, Italy (Member of CRUST - Centro inteRUniversitario per l'analisi SismoTettonica tridimensionale con applicazioni territoriali)
[4]EarthScope Consortium, Boulder, CO, 80301, USA

*Correspondence to*: Paola Montone (paola.montone@ingv.it)

**Abstract.** The ICDP STAR drilling project aims to study the seismic and aseismic fault slip behaviour of the active low-angle Alto Tiberina normal Fault (ATF) in the Northern Apennines, Central Italy, drilling and instrumenting six shallow boreholes with seismometers and strainmeters. During the STAR field work, a geophysical downhole logging campaign was carried on defining the optimal target depth for instrument deployment and formation rock characterization. In particular, the main objectives of this study were to define in situ physical properties of the rocks and the tectonic discontinuity geometry along the boreholes. The downhole logging data provide new findings and knowledge especially with regards to the physical properties such as resistivity, gamma ray and wave velocity. The collected parameters were compared to the results of literature data collected in similar lithologies, as well as with the results of logging performed in deeper wells drilled for commercial purposes. The physical properties of the Mesozoic-Early Tertiary calcareous formations show low Gamma Ray values and high compressional (Vp) and shear wave (Vs) velocities (up to 5.3 km/s and 2.9 km/s, respectively), whereas the overlying clay-rich Late Tertiary formations exhibit high Gamma Ray and low resistivity and relatively low Vp and Vs values (up to 3.5 km/s and 2.0 km/s, respectively). The results obtained from the analysis of the orientations of the tectonic structures, measured along the six boreholes, show a good agreement with the orientations of the present-day extensional stress field, NE-SW oriented. Our study allowed to bridge the gap between the physical properties obtained from literature data and those obtained from the deep wells measurements, representing a possible case history for future projects. These new data will contribute to the advancement of knowledge of the physical properties of the rocks at shallow depths, typically overlooked.



## 1 Introduction

The aim of the STAR drilling project (A Strainmeter Array Along the Alto Tiberina Fault System) is to study seismic and aseismic slip on active high- and low angle seismogenic normal faults (Chiaraluce et al., 2023) in Central Italy, an area affected by seismic events with magnitude up to Mw 6.6 (Fig. 1). The STAR drilling project is an international effort contributing to the infrastructural implementation of the Alto Tiberina Near Fault Observatory (TABOO-NFO) (Chiaraluce et al., 2014a, 2014b; Chiaraluce et al., 2022), a long-term research infrastructure mapped by the EPOS initiative as one of the European Solid Earth Science facilities providing open access data to the international community (http://www.epos-eu.org). STAR is one of the International Continental Scientific Drilling Program projects with a primary focus on long-term borehole monitoring of fault-zone deformation (e.g., Bohnhoff et al., 2017; Fischer et al., 2022).



**Figure 1: Seismicity in the study area and main seismic sequences in central Italy in the last 45 years. Legend: From 1 to 6: 1- Norcia 1979 (Deschamps et al., 1984); 2- Gubbio 1984 (Haessler et al., 1988); 3- Colfiorito 1997 (Chiaraluce et al., 2003); 4- Gualdo Tadino 1998 (Ciaccio et al., 2005); 5- L'Aquila 2009 (Valoroso et al., 2013); 6- Central Italy 2016-17 (Michele et al., 2020); 7- seismicity in the period 2010-2014 (Valoroso et al., 2017); 8 and 9- location of main events of the sequences and related focal mechanisms (Italian CMT dataset; Pondrelli and Salimbeni, 2006; Pondrelli et al., 2006; Quick Regional Moment Tensors); 10- STAR boreholes (TSM). The red box is the area of Figure 2. In the lower left corner, a cross section of the seismicity (Michele et al., 2020).**



46

To improve the comprehension of the processes controlling fault mechanics and earthquake generation, the STAR drilling project installed short period (2Hz) seismometers (three-component borehole geophones) and strainmeters (Gladwin Tensor Strainmeters, GTSM) in six shallow boreholes (maximum depth 160 m) purpose-drilled. The seismometers were installed to monitor and record seismicity of the low-angle Alto Tiberina normal Fault (ATF) and its main antithetic splay, the Gubbio normal fault (Mirabella et al., 2004; Caricchi et al., 2015) (Fig. 2). Borehole strainmeters were deployed because they are the only instruments able to measure small creep events, as demonstrated in similar experiments focused on other faults, such as the creeping section of the strike-slip San Andreas fault near Parkfield (Langbein et al., 2006). Seismic and strain observations from the STAR boreholes monitoring will be integrated with regional observations on active seismicity, on deep crustal structure and on the present-day stress field.

The six 80-160 m shallow monitoring boreholes, named TSM1-6, (see locations in Figure 1) were drilled surrounding the creeping portion of the ATF in two phases: during the Fall of 2021 and Spring of 2022. The STAR drilling operations were supported by the acquisition in all the boreholes of a wide set of geophysical logs including: optical (OBI), acoustic (ABI), caliper (CAL), gamma ray (GR), fluid temperature conductivity (FTC), sonic (FWS), resistivity and spontaneous potential (ELOG) logs. The borehole geophysical measurement purposes for the STAR drilling project were twofold: firstly, to characterise the physical properties of the rock formation in the subsurface and, secondly, to identify an intact and competent fracture-free interval in each borehole in which to deploy the strainmeter and seismometer. After the completion, each borehole was instrumented and ready for data acquisition (Chiaraluce et al., 2023).





**Figure 2: Geological setting of the study area. The location of the six STAR boreholes (red circles) drilled in Fall 2021 (TSM01, 02, and 03) and in Summer 2022 (TSM04, 05, and 06), and two deep commercial boreholes (yellow circles), San Donato 1 and Mt. Civitello 1, are displayed. Modified from (Mirabella et al., 2011). The geological cross-sections A-A' and B-B' are in Fig. 3.**

The objective of this paper is to provide a critical overview of the physical properties of the in-situ rock formations and their fracture characteristics based on analysis and interpretation of downhole logging data. Particular attention was paid to optical and acoustic image logs with the aim of identifying intact rock and structural discontinuities. In fractured rock masses, discontinuities have significant control over the rock mass behaviour. Mapping at depth the fractures and their geometry helped us identify optimal intervals to host seismometers and strainmeters. In order to work properly and obtain reliable data, these



instruments must have a perfect coupling with the rock mass: therefore, borehole seismic installations have to take into account
the borehole diameter and tilt, temperature profile, lithology and fracture distribution.
In this paper, after a brief description of the seismicity of central Italy and a geological and tectonic overview of the area where
the boreholes are located, we describe the main results of the operated logging. Geophysical downhole measurements provide
a contribution to better define the physical properties of the Umbria-Marche carbonate multilayer (mainly limestones and
marls) and of the overlying turbidites, cropping out in this area. These results are then compared with the analogue
measurements, acquired in much deeper boreholes, drilled in the same region for hydrocarbon exploration purposes, as well
as with the available, recently acquired laboratory measurements (e.g. De Paola et al., 2009; Smeraglia et al., 2014; Trippetta
et al., 2010, 2021) giving food for thought about the effects of the confining pressure on the physical parameters of the rocks.
In particular, the results related to the P-wave velocities obtained from the sonic log readings along the six STAR boreholes
have been compared with the previous results related to the same geological formations (e.g. Barchi et al., 1998; Diaferia et
al., 2006; Bigi et al., 2011; Mirabella et al., 2011; Scisciani et al., 2014; Porreca et al., 2018; Montone and Mariucci, 2020;
Trippetta et al., 2021). From the geophysical log analysis, we have defined and characterised several planar discontinuities
along each borehole, related either to primary (bedding) or to secondary (tectonic) structures (fault and/or fractures). The
orientations of the tectonic structures, recognized along the boreholes, have been considered together with other available data
and compared with the present-day stress field.
Summarising, our paper aims to bridge the gap between the physical properties obtained from literature data (e.g. laboratory
analyses) on outcrop samples and those obtained from the wellbore measurements of oil and gas companies (such as AGIP,
ENI; https://www.videpi.com/videpi/pozzi/consultabili.asp), which investigate significantly greater depths. The new data of
this study will contribute to the advancement of knowledge of the physical properties of the rock masses at relatively shallow
depths (0-200 m), typically overlooked. Overall, also considering the results obtained in the analysis and interpretation of the
data in this study, the outcomes from the STAR drilling project will provide a better understanding of the behaviour of the
main active faults, addressing fundamental questions about the relationship between creep, slow slip, dynamic earthquake
rupture, and tectonic faulting (Chiaraluce et al., 2023).

## 2 Seismotectonic and geological framework of the area

Seismicity in Central Italy is mainly characterised by shallow crustal earthquakes (5–15 km depth) localised along the
Apennine belt with maximum magnitudes of about 6.6 (Chiaraluce et al., 2017a; Chiarabba et al., 2005). Earthquake focal
mechanisms show a prevalent normal faulting regime, with a NE–SW striking extension, consistent with other data
characterising the active stress field in this area, such as breakouts and active faults (Mariucci and Montone, 2024).
In the last 45 years, Central Italy has experienced several crustal normal-faulting earthquakes (Fig. 1) causing surface faulting
as well, visible fractures and significant damage (Cinti et al., 1999; Barchi and Mirabella, 2009; Boncio et al., 2010; Emergeo



working Group, 2010; Pizzi et al., 2017; Villani et al., 2018; Barchi and Collettini, 2019). The most significant earthquakes
occurred in the past, with a moment magnitude greater than 5.5 (Fig. 1), are the Mw 5.8 Norcia in 1979, the Mw 5.6 Gubbio
in 1984 and the seismic sequence of Colfiorito-Gualdo Tadino in 1997-98, with the largest event Mw 6.0 (Cello et al., 1997;
Amato et al., 1998; Amato and Cocco, 2000; Boncio and Lavecchia, 2000; Ciaccio et al., 2005; Mildon et al., 2016). Finally,
the seismic sequence that began in 2016 in Amatrice (Tinti et al., 2016; Chiaraluce et al., 2017b; Chiarabba et al., 2018)
occurred with three main events (Mw 6.2 in Amatrice, Mw 6.1 in Visso, and Mw 6.6 in Norcia), causing about 300 deaths,
injuries and the destruction of numerous historic centres (Fig. 1).
The six boreholes of the STAR drilling project were drilled in the NW part of the actively extending area of the Umbria-
Marche Apennines (Fig. 2), a NE-verging, arc-shaped foreland fold-and-thrust belt, representing the eastern part of the
Northern Apennines of Italy. Within the study area the compressional structures (folds and thrusts) are mostly arc-shaped with
a roughly NNW-SSE trend and were formed in Late Miocene (Tortonian-Messinian age). They affect a pre-orogenic Jurassic-
Paleogene carbonate multilayer (Umbria-Marche succession) (e.g. Cresta et al., 1989), overlain by a thick succession of syn-
orogenic Neogene turbidites, marls and sandstones, deposited in the Northern Apennines foreland basin (e.g. Barchi, 2010).
The compressional structures are cut and displaced by later (Late Pliocene-Quaternary) NW-SE striking normal faults, which
are related to an extensional stress field oriented in a NE-direction, responsible for the present-day seismicity of the region.
The normal faults attitude is consistent with the extensional stress regime inferred from earthquake focal mechanisms and
borehole breakouts (Mariucci et al., 2008; Montone and Mariucci, 2016; Villani et al., 2018).
The most prominent normal fault exposed in the study area is the SW-dipping Gubbio fault, down-throwing the western
backlimb of the Gubbio anticline (Fig. 3). The Gubbio fault is antithetic to a major NE-dipping extensional detachment, i.e.,
the ATF (e.g. Mirabella et al., 2011; Lavecchia et al., 2017). The fault dip of less than 30° makes the ATF an unfavourably
oriented geological structure for reactivation with respect to the regional stress field. The ATF and its high-angle antithetic
splays release continuous microseismicity, and rarer moderate sequences, e.g. in 1984 (Haessler et al., 1988) and in 2010-2014
(Marzorati et al., 2014).
The stratigraphy of the study area, from top to bottom, can be summarised as follows (Barchi, 2010): i) marine and continental
Plio-Quaternary sediments, mainly clays and sands in different combinations and with different degrees of compaction; ii) a
thick Neogene synorogenic turbidite succession, namely Marnoso-Arenacea Fm. (Miocene), formed of alternated shales and
sandstones, with strong vertical and lateral variability; iii) a hemipelagic marly succession (Schlier, Bisciaro and Scaglia
Cinerea Fms.; Eocene -Miocene); iv) a carbonate pelagic sedimentary sequence of the Umbria–Marche domain (Mesozoic–
Paleogene) that includes not only limestone and chert-bearing limestone but also marl and clay (Scaglia Variegata, Scaglia
Rossa, Scaglia Bianca, Marne a Fucoidi and Maiolica Fms.); v) Lower Jurassic massive platform carbonate (Calcare Massiccio
Fm.); vi) Upper Triassic evaporitic succession, consisting of alternated anhydrites and dolostones (Anidriti di Burano Fm.);
vii) Middle Triassic and/or older continental and shallow marine meta-sediments (Verrucano Fm. s.l.).



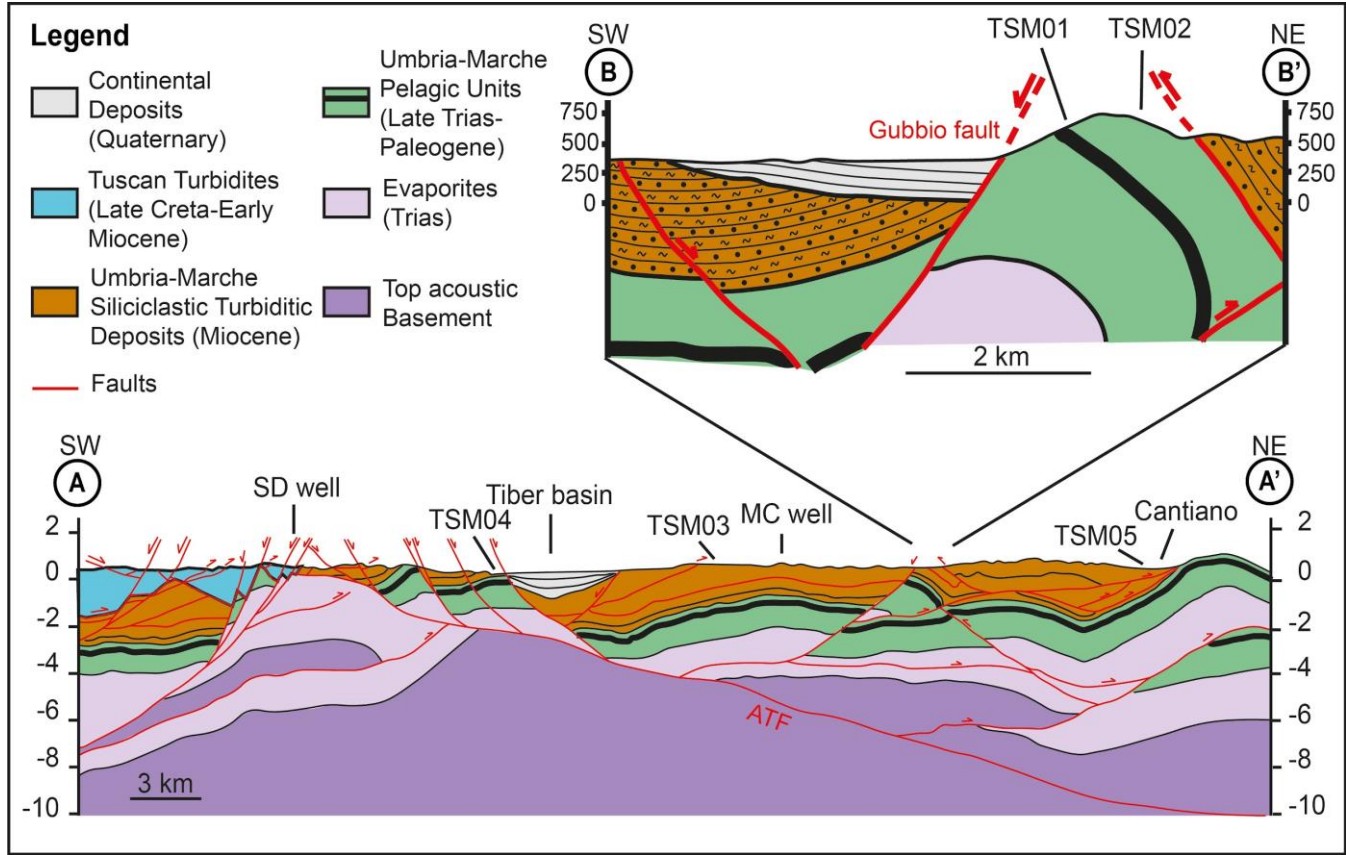

**Figure 3: Geological cross-sections (A-A' and B-B' in Figure 2). The geometry at depth of the main tectonic structures such as the Alto Tiberina low angle normal fault (ATF) and the Gubbio fault (Mirabella et al., 2011) are shown. SD well, San Donato 1 well; MC well, Mt. Civitello 1 well.**

Within the STAR study area (Figures 2 and 3), two deep wells (San Donato 1 and Mt. Civitello 1) were drilled in the past by Italian oil Companies. The wells stratigraphy is schematically reported in previous literature (e.g. Mariucci et al., 2008; Mirabella et al., 2011; Caricchi et al., 2015). The San Donato 1 well (SD), reaching a depth of 4763 m, was drilled by SNIA-BPD in 1983-84 and is located approximately 20 km southwest of the Mt. Civitello 1 well (MC). The SD well is situated very close to the ATF, intersecting it at a depth of 326 m, where the Miocene Marnoso-Arenacea turbidites directly overlie the Triassic evaporitic succession, extending down to a depth of about 3000 m. At greater depth, the well penetrates the metamorphic acoustic basement in tectonic contact with the evaporitic succession (Fig. 3). The MC well, drilled by AGIP in 1988-89, located near the Gubbio fault, reaches a depth of 5600 m. From the surface to a depth of about 1000 m, the well crosses the turbidite succession. Then the well passes through the carbonate Meso-Cenozoic pelagic sequence. From about 2800 m to the bottom at 5600 m, the well crosses the Triassic evaporitic succession.



## 3 Method: downhole logging processing and analysis

The knowledge of the petrophysical properties of the different litho-stratigraphic units is an important aspect to understand the subsurface. In the lack of coring material, as in the STAR project, data acquisition from downhole logging becomes the key element in determining the rocks physical characterization (i.e. Rider and Kennedy, 2011). Downhole logging is a method to gain continuous, in situ high-resolution data of various physical or structural rock parameters collected within a borehole. For the STAR drilling project, the downhole logging was performed for all six boreholes in order to allow detailed sedimentary facies assessment and to identify the best location to deploy the strainmeters and seismometers at depth. All six boreholes were logged by slimhole sondes (for details see Table S1 and text in the Supplementary), and following the standard methods in this field (Serra, 1984; Ellis and Singer, 2007; Rider and Kennedy, 2011; Schön, 2015; Pierdominici and Kück, 2021). Downhole measurements were conducted in each borehole after the drilling operations and executed mainly in the open hole (OH) sections, only GR ran also in the cased section (CH). The following downhole measurements were successfully recorded: total gamma ray (GR), full waveform sonic (Vp and Vs), temperature (T) and conductibility (COND), three-arm caliper (CAL), resistivity (RESIST), single point resistance (SPR), and acoustic (ABI) and optical images (OBI). We have summarised the logging measurements and logged interval for each borehole in Table S1. For the drilling operations, water was used as drilling fluid allowing to run the OBI. Borehole quality has been determined by vertical and horizontal deviation of the borehole and the condition of the borehole wall. In order to pursue the objective of the STAR drilling project and for proper use and performance of the instruments, all six boreholes were drilled within 5° from the vertical. The deviation of each borehole was then checked as part of the logging program. The boreholes have an inclination of less than 2° except for the borehole TSM06 where the inclination is between 4.3° and 5.1°. Based on i) the smooth borehole wall, ii) log without intervals of large washouts and iii) internal consistency for several tools (i.e., three different types of borehole diameter measurement), we infer that the log quality and reliability are very good for almost all sondes over the entire length of each hole.

Below, we have summarised the main scientific purpose of each sonde.

The total gamma ray log (GR) measures the natural radioactivity of the rock. The GR comes from the radioactive isotopes of potassium (40K), uranium (238U decay series) and thorium (232Th decay series). Potassium is found primarily in clay minerals, micas, and potassium feldspar; thorium is commonly associated with clay minerals and volcanic ash layers; uranium is found in heavy minerals, glauconite and organic rich intervals and may be bound to clay. Relatively high values in GR log are often associated with the influx of clay and coarser materials, while relatively low GR values generally indicate sedimentation of biogenic carbonate, organic carbon, or silica (e.g. Rider and Kennedy, 2011). We performed the GR log to detect layers of clay and to identify changes in lithology. GR was also recorded in drill pipes of all six STAR boreholes, although the signal appears a bit dampened. In essence, GR log characterises the different lithology crossed by the borehole allowing to identify layers (thickness and lithotype) through the different clay content.

A three-arm caliper (CAL) sonde was used to measure the borehole diameter and to determine how smooth the borehole walls are. The strainmeter requires relatively smooth walls with no blowouts or fractures.



The fluid temperature-conductivity (FTS) measures the temporary temperature and conductivity of the borehole fluid. Both
parameters can show strong variations caused by drilling activities inside the hole but also can detect flow of fluids into or out
of the formation. These logs are good indicators of areas of active flow or open fractures, therefore are used to exclude areas
or intervals with major fractures that would have affected both instrument placement and data quality.
The ELOG sonde - including RES and SPR - measures the rock capability to conduct electric currents. The tool provides
resistivity profiles with four different depths of investigation. This measurement provides information about permeability,
porosity, water types and geological formation properties. In particular, in massive rocks with very low matrix
porosity/permeability, the resistivity logs identify fluid filled fracture zones (fracture permeability). So, we ran this logging
sonde for the same reason as the FTS. Only for the TSM01 and TSM02 boreholes the ELOG logs were not acquired.
The Full Waveform Sonic (FWS) sonde measures the velocity of sound waves through the rocks, which varies depending on
lithology, rock texture and porosity. The sonic velocity measurement is used for identification of compaction of lithologies,
facies recognition and fracture identification. The velocity has been determined by measuring the travel time of sonic pulses
between transmitters and four receivers. We have reprocessed the raw sonic waveforms to estimate the P and S-wave velocities
using a combination of first arrival trace picking for P and S waves, along with additional semblance analysis.
Borehole image sondes provide a continuous oriented high resolution image of the borehole walls. Images collected in the
STAR project have been obtained from acoustic (ABI) and optical (OBI) tools. The latter acquires a true-colour optical image
of the borehole wall, and the acquired data are displayed in one oriented unwrapped image. The sonde operates only in a
transparent drilling fluid like fresh water or air. For TSM04 and TSM06 boreholes no optical image is available due to the
high turbidity of the drilling water. The acoustic image data are visualised as two 360 degrees north-oriented images (travel
time and amplitude) of the borehole wall versus depth. The travel time (TT) provides information about the borehole shape
and the acoustic amplitude (AMPL) depends on the roughness and shape of the borehole wall and its acoustic properties, which
depend on variations with texture, mineralogy, compaction, and fracturing (e.g. Davatzes and Hickman, 2010). These AMPL
and TT images are visualised in colours based on their value range. Here, in the AMPL image, strong contrast (high amplitude,
bright colour) indicates a strong signal and good reflection, and low contrast (low amplitude, dark colour) indicates weak to
missing signals (scattered or absorbed impulse). In the TT image, the bright colours indicate a short time period (fast) for the
impulse to go from the transducer and back to the receiver; the dark colours represent a long time period (slow), which means
widened size of the borehole (e.g. Pierdominici et al., 2020). Each planar discontinuity, such as fracture, fault, bedding, appears
in the images as sine waves (e.g. Davatzes and Hickman, 2010). To obtain a correct geometry of the planar structures (dip
azimuth and dip) the TT and AMPL images have been prior corrected by diameter, inclination and orientation of the borehole.
We have grouped the features in six categories (see Section 4.3): open (in red), filled (in grey), bedding (in green), stylolite (in
turquoise), cherty layer (in dark grey) and weak zones. The acoustic imager tool alone cannot distinguish between open or
closed/filled structures (used here in a general term, i.e., including faults, fractures, and veins). Based on the comparison of
these two images, we might distinguish so-called "open" structures based on their contrast of ultrasonic AMPL and the





corresponding response in the TT. The structures defined as "closed" are visible only in the amplitude image. To enhance the
travel time and amplitude images, static and dynamic (10 cm vertical window) normalizations were applied. Raw, static,
dynamic images display minor differences of each other in the resulting images depending on the scale and variations between
different intervals and features. The displayed images here refer to the static normalisation. The ABI and OBI were also used
to determine the borehole trajectory based on borehole's deviation from vertical (DEVI) and the direction of this deviation
with respect to magnetic north (hole- or drift-azimuth; HAZI). For the installation of strainmeters and seismometers, all STAR
boreholes had to be very close to the vertical. Thus, this type of acquisition played a key role in knowing the condition of the
borehole and decreeing the successful installation of the instruments.

## 4 Data and Results

### 4.1 Borehole description

The boreholes are located within an area of about 1500 km$^2$, centred around the town of Gubbio (Fig. 2). Among the six new
boreholes, TSM01 and TSM02 were drilled across the late Mesozoic-Early Tertiary carbonates, cropping out along the crest
of the Gubbio anticline, which represents the maximum structural culmination of the study area (Fig. 4 a, b). The other
boreholes were drilled in the Tertiary marls and sandstones, cropping out in the northern part of the study area, to the W
(TSM04), N (TSM03 and TSM06) and NE (TSM05) respect to the Gubbio anticline axis (Fig. 4 c, d, e and f).
See supplementary files for a lithological detailed description (Supplementary 1).





**Figure 4: Geological cross-sections across the STAR boreholes. Different members of Marnoso-Arenacea Fm. are shown (FMA1, 2, 3, 4, 9 and MUM).**

## 4.2 Log results

The downhole logging data are described and interpreted to provide new findings and knowledge to complement those obtained from similar rock types sampled in the field outcrops (Fig. 5 and Figs. 6-9).

Results from the downhole logging analysis are summarised in Table 1 and Table S1 displaying the average values and the related standard deviation. More details are described in Supplementary 2.



| Borehole | ICDP-ID | Depth | | GR | | Vp | Vs | RES | SPR | T | COND |
|---|---|---|---|---|---|---|---|---|---|---|---|
| | | m | | cps | | m/s | m/s | Ωm | Ωm | °C | μS/cm |
| | | CH | OH | CH | OH | | | | | | |
| TSM01 | 5070_1_A | 0.0-99.3 | 99.3-133.0 | 17.9±13.3 | 8.4±4.4 | 5324±371 | 2885±112 | n.d | n.d | 12.3±0.6 | 380.9±39.9 |
| TSM02 | 5070_2_A | 0.0-97.0 | 97.0-160.3 | 11.7±5.4 | 13.4±5.6 | 4867±184 | 2505±500 | n.d | n.d | 14.7±1.3 | 456.4±61.5 |
| TSM03 | 5070_3_B | 0.0-32.0 | 32.0-80.0 | 61.5±12.7 | 104.2±15.9 | 3204±168 | 1800±130 | 29.2±4.7 | 104.3±10.4 | 13.6±0.2 | 451.6±7.4 |
| TSM04 | 5070_4_A | 0.0-79.3 | 79.3-101.0 | 61.5±10.0 | 85.1±13.2 | 2972±207 | 1993±118 | 8.6±1.0 | 42.2±4.6 | 16.2±0.3 | 694.3±91.7 |
| TSM05 | 5070_5_A | 0.0-82.3 | 82.3-118.0 | 48.6±8.7 | 81.2±9.3 | 3451±160 | 1879±148 | 11.5±3.2 | 53.4±7.2 | 16.5±0.7 | 634.0±8.2 |
| TSM06 | 5070_6_A | 0.0-80.0 | 80.0-117.5 | 62.4±9.2 | 99.7±9.35 | 3422±306 | 1896±119 | 18.2±3.2 | 76.1±8.9 | 18.2±0.03 | 487.3±34.4 |

**Table 1: Geophysical properties of the rocks for TSM boreholes. GR: gamma ray (count per second); Vp and Vs: P- and S wave velocity, respectively; RES: resistivity; SPR: single point resistance; T: temperature; COND: conductivity; CH: cased hole; OH: open hole.**

GR log response (open and cased hole) in TSM01 and TSM02 boreholes is generally very low (Fig. 5, Table 1). In TSM01, the low GR, associated with the limestone Scaglia Bianca Fm., slowly increases in the marly Marne a Fucoidi Fm., proportionally to the enrichment of the clay component as well evidenced in the interval between 55.0 m and 66.5 m. The sharp contact between the Marne a Fucoidi Fm. and the underlying limestone Maiolica Fm. is well indicated by the dramatic decrease in GR values (from 50 cps of Marne a Fucoidi Fm. to 14 cps of Maiolica Fm.). The open-hole section interests only the Maiolica Fm. with an average GR value of 8.4 cps. In TSM02, GR records a very low response related to the three pelagic formations intersected, predominantly consisting of limestones, which are interbedded with sporadic, relatively thin marly layers as recorded by the increase in GR values (e.g. 73-75 m, 102-104 m, 134-135 m in Scaglia Rossa Fm.). In the TSM03, TSM04, TSM05 and TSM06 boreholes, GR log shows higher values which vary between 49 cps in TSM05, and 104 cps (open hole) in TSM03. The GR values are lower in the cased section because the signal is damped by casing. In the open-hole section, on the other hand, the GR response for all the four boreholes is generally high, displaying relatively uniform cps value ranging between 81 and 104. Intervals with low GR are few and restricted and likely associated with the presence of fractures and/or thin sandstone layers. The highest GR values are almost exclusively associated with the marly layers that dominate the Marnoso-Arenacea Fm. (TSM03, TSM04 and TSM06) and the Schlier Fm. (TSM05).









**Figure 5: STAR boreholes with lithostratigraphic profile and gamma-ray log. Each borehole has a conductor casing for the first 9 m, followed by a casing and an open-hole section. ID: inner diameters. Only the total gamma (GR, here shown) was also run through the casing, while all other measurements were performed only in the open hole. Seismometers and strainmeters were deployed at the bottom of the open section of each borehole (for details see Fig. S1).**

P (Vp) and S (Vs) wave velocities obtained from the full wave sonic log were measured in the open–hole section down to the bottom of the hole showing a wide range of values between different boreholes: Vp varies between 2972 and 5324 m/s and Vs between 1800 and 2884 m/s (Table 1; Figs. 6-9). Higher values were recorded along the boreholes that intersected more competent lithologies, especially limestones as Maiolica Fm. in TSM01 and Scaglia Rossa in TSM02. A significant decrease of both Vp and Vs was detected at open fractures occurrence. The dynamic Poisson ratio was computed using the specific formula from shear and compressional sonic logs to compare it to the fracture porosity. As expected, rocks with a low Poisson's ratio show a higher fracture density (Figs. 6-9).

Resistivity was measured in TSM03, TSM04, TSM05 and TSM06 boreholes. The values vary from 9 to 29 Ωm according to the response of sonic logs and fracture presence (Table 1); however, with the same lithology the resistivity values vary following the variations of single point resistance (SPR).

Temperature measured in the boreholes (Table 1) is quite constant within a range of 12 to 18°C reflecting the borehole fluid rather than the "formation temperature". Due to the shallow investigated depth, the results are not very significant. See Section 3 for details.

Conductivity of the drilling fluid is directly proportional to the concentration of dissolved minerals and thus to its salinity: the highest values (694 μS/cm) were found in the TSM04 borehole near the area with intense CO2 emissions of deep origin (Table 1).

### 4.3 Structure analysis

In this section we present the analysis of optical and acoustic images performed to identify the main discontinuities along the boreholes (Figs. 6-9). As mentioned before, we have grouped the features in six categories: open (in red), filled (in grey), bedding (in green), stylolite (in turquoise), chert layer (in dark grey) and weak zones. The filled and open fractures usually have a thickness from a few mm up to 1-2 cm. Furthermore, we plotted the main tectonic structures (open and filled fractures and weak zones) as rose diagrams splitting the data according to their dipping (> and < 45°).

**TSM01**. The analysis of OBI and ABI images allowed us to identify and detect the main discontinuities crossed by the borehole (Fig. 6). The sub-horizontal discontinuities correspond to bedding planes, while the filled discontinuities can be interpreted as later-filled fractures or cleavage planes. Moreover, numerous stylolites and chert layers are present within the Maiolica Fm., both parallel to bedding. A detail of the main features is shown in the inset of Fig. 6-i. A total of 69 discontinuities (only open and filled fractures) were recorded along the open-hole section showing a homogeneous distribution along the borehole with a maximum of 8 structures per metre. The preferential orientation is NW-SE, corresponding to a dip azimuth of N200° (Fig.





298    10). The dip of these planes is generally low (around 25°), contributing to the azimuthal dispersion of the data. However, some

299    discontinuities with steeper dips (ranging from 60° to 80°) are still NW-SE oriented.







**Figure 6: Downhole logging measurements performed along the TSM01 borehole (Maiolica Fm.). At the bottom: a detail of the structures intersected by the borehole.**

**TSM02**. The OBI and ABI images available for structural analysis investigated two different depth ranges. The OBI was performed from 96.2 to 147.0 m and the ABI from 144.2 to 159.6 m. The OBI image clearly shows continuous and parallel layering and especially the interlayering of thin clay layers in the limestone. The Scaglia Rossa Fm. is also affected by numerous thick filled and open fractures. The analysis of the planar discontinuities identified on both images allowed us to distinguish bedding planes, filled and open fractures (Fig. 7). The latter two structures (195 counts) revealed a predominant orientation of NNW-SSE, corresponding to a dip azimuth of N204° (Fig. 10), with an average structure frequency of 6 per metre and a high discontinuities concentration around 120 to 125 m (Fig. 7-i). The majority of open and filled fractures have dipping values higher than 45°, the open ones dipping almost exclusively to NE, and the closed ones to SW (Fig. 10).







**Figure 7: Downhole logging measurements performed along the TSM02 borehole (Scaglia Rossa Fm.). At the bottom: a detail of the structures intersected by the borehole.**

**TSM03**. The poor quality of the OBI and ABI images made it difficult to perform a successful structural analysis. The unclear images are related to the high turbidity of the fluid in the borehole due to non-flushing of the borehole prior to acquisition and the high speed of running the OBI and ABI probes. Only the bedding planes were recognized showing a mean orientation of NW-SE (dip azimuth of N201°) with a very low dip of 11°. Owing to the low quality of the images, a geological survey was performed to measure bedding and main tectonic structures directly on the Marnoso-Arenacea outcrop near the drilling site (Fig. S2). Bedding varies from N204 near the borehole, to N167 and N155, dipping about 5-15°W, consistent with what was observed from the data analysis along the borehole. The thickness of the sandstone levels of Marnoso-Arenacea Fm. is up to 100-120 cm, while the thickness of the grey marly layers ranges from 1 to 10 cm; they are laminated with cleavage planes sub-parallel to the bedding. There are fractures and sometimes sub-vertical faults, which are clearly visible in the sandstone layers, with an average fracture orientation of N050 sub-vertical, N215 sub-vertical faults with extensional displacement, N213 dipping 76°W with left-lateral striae overlapped with oblique striae (pitch 55°, Fig. S2).

**TSM04**. The borehole intersected numerous discontinuity planes along the entire length of the measured log (Fig. 8). Although the log length is approximately only 20 m (79.8 to 100 m), the quality of the ABI images is significantly better compared to the TSM03 and TSM06 boreholes drilled in the same formation (Marnoso-Arenacea Fm.). The drastic decrease of cloudy fluid in the borehole is the combined result of wellbore flushing operations before starting log acquisition, and the low running speed of the ABI probe resulting in a good ABI image quality. 195 discontinuities between filled and open have been detected on image log with an average orientation of N184 and dipping never exceeding 60° (Fig. 10). In the field, upstream of the drilling site (Fig. S2), it was possible to measure bedding that is N180 oriented, dipping 25°W related to a small Late Cretaceous-Early Miocene outcrop of Tuscan turbidites (Fig. 2).







**Figure 8: Downhole logging measurements performed along the TSM04 borehole (Marnoso-Arenacea Fm.). At the bottom: a detail of the structures intersected by the borehole.**

**TSM05**. Analysis on OBI and ABI images allowed to clearly identify tectonic structures such as open and filled structures (Fig. 9). The bedding planes instead are dubious and difficult to recognize. Discontinuity planes show a spacing quite dense (12 planes per 1 m) up to 94.8 metres; below, they are very sparse, with sections of up to 3 m without discontinuity probably also due to the acquisition and to the presence of cloudy drilling fluid in the borehole. From the image analysis a total of 114 planar structures have been identified with very consistent NW-SE orientation (corresponding to a dip azimuth of N219; Fig. 10). From this dataset, we have marked at least 4 different categories of discontinuity according to their dip, the presence or absence of filling and their aperture. A first category of discontinuities is characterised by dip greater than 45° which is related to both open and filled fractures. They show planes approximately NW-SE oriented with dip both towards SW and towards NE. The filled fractures with dip less than 45° show a similar NW-SE trend as well as the open fracture zones including wider zones (decimetre thicknesses up to 1 m; Fig. 9-i) still have a NW-SE orientation and dips ranging from very low to almost vertical. The bedding planes are mainly sub-horizontal.



351



**Figure 9: Downhole logging measurements performed along the TSM05 borehole (Schlier Fm.). At the bottom: a detail of the structures intersected by the borehole.**

**TSM06**. The structural analysis has been performed only on ABI image log allowing to identify 68 planar structures of which 27 open and 41 filled in the Marnoso-Arenacea Fm. (Fig. 10). The planar discontinuities show a prevailing NE-SW orientation, and dip 20-30°W. There is also a minor NNW-SSE oriented data concentration characterised by steeper dip up to 60°NE. A weak zone oriented N300 dipping 54°NE with a width of about 21 cm, has also been identified along the TSM06 borehole. In the field very close to the drilling site, bedding is N190 oriented, dipping 30°W. In the surroundings, low angle bedding planes are also NW and NE oriented. Fracture planes are N125 oriented, dipping 88°S close to the drill; other high angle tectonic structures are also E-W and NE-SW (Fig. S2).






**Figure 10: Rose diagrams and polar projections related to open and filled structures detected along the STAR boreholes (except TSM03). N: number of structures; DIP: poles of the planes (lower hemisphere) with the contouring (in percentage). The last column represents the fractures dipping >45°.**


## 5 Discussion and Conclusion

In the framework of the STAR project, six shallow boreholes were drilled, conducting geophysical downhole logs accurately, aimed to identify the most suitable depth in each borehole for the deployment of seismometers and strainmeters. The results





of the seismometer and strainmeter network will require more time to obtain useful information for the project itself, dedicated
to the study of the brittle upper crust, and the structure and behaviour of the Alto Tiberina low-angle seismogenic normal fault,
in particular. This paper pertains solely to the initial phase of the STAR project, specifically focusing on the analysis of
downhole geophysical well logs.
Downhole logging was the only way to characterise the borehole section, allowing the physical and structural properties of
each geological formation to be determined, due to the lack of core samples. These in situ measurements are sensitive to
formation properties on a scale that is intermediate between those obtained from literature data analysis performed on core or
outcrop samples and deep geophysical measurements, performed by exploration and production drilling companies.

## 5.1 Physical properties


Regarding the calcareous formations of Maiolica and Scaglia Rossa Fms., crossed by the boreholes TSM01 and TSM02,
respectively, the physical properties of these competent rock types are well reflected by the acquired log data (Table 1),
showing low average GR values (less than 18 cps) and relatively high average values of Vp (5.3 and 4.9 km/s, respectively)
and Vs (2.9 and 2.5 km/s, respectively). The almost pure limestones of the Maiolica Fm. are characterised by lower GR and
higher Vp and Vs values, in comparison with the Scaglia Rossa Fm., where the clay content is significantly higher (up to 20%
in the Tertiary upper portion of the Scaglia Fm., e.g. Arthur and Fischer, 1977).
The TSM03, TSM04, TSM05 and TSM06 boreholes were drilled in the marly intervals of the Neogene successions of the
Marnoso-Arenacea Fm. and Schlier Fm. (TSM05). These clay-rich rocks are coherently characterised by high average GR
(between 81 cps in the Schlier Fm. and 104 cps in the Marnoso-Arenacea Fm., TSM03) and low RES (between 9 Ωm and 29
Ωm). The measurements include lower values for TSM04 (9 Ωm) and TSM05 (12 Ωm), Marnoso-Arenacea and Schlier
respectively, intermediate values for TSM06 (18 Ωm), and higher values (up to 29 Ωm) for TSM03.
In greater detail, the average values are not totally representative of these complex formations, typically consisting of alternated
marls and sandstones: similar suggestion derives from the relatively low Vp (3.0 to 3.5 km/s) and Vs (1.8 to ~2.0 km/s) values.
The temperature recorded in the boreholes (Table 1) is rather constant with values between 12°C and 18°C without showing
any significant variations related to e.g. outflow or inflow zones. Unfortunately, the shallow depth of the investigation limits
the significance of the results.
We compared our RES results with two deep well data (San Donato 1 and  Mt. Civitello 1 wells; https://www.videpi.com)
drilled in the same formations and considering the same analysed depth interval, showing similar values between 20 Ωm and
30 Ωm, for the Marnoso-Arenacea Fm. (Fig. S3). We have also investigated the resistivity of the Schlier Fm. along the Canopo
1 well (https://www.videpi.com), although this latter is located far away from the study area, approximately 80 km NNE of
Gubbio. Also in this case the resistivity values, around 7 Ωm, are comparable with our data (Fig. S3).





Information obtained from the velocity logs are more significant, including all 6 boreholes. We compared the Vp values
recorded in the six shallow (depth < 0.2 km) STAR boreholes (Table 1) with the data derived from sonic log analysis in much
deeper wells (depth > 4 km), drilled in the same region for industrial purposes, and recently analysed by Montone and Mariucci
(2020) (see also references therein) and Trippetta et al. (2010). These authors report average values between 4 and 4.8 km/s
for Marnoso-Arenacea and Schlier Fms., 5.8 km/s for Scaglia Rossa Fm. and between 5.9 and 6.2 km/s for Maiolica Fm. All
these values are about 30 % higher with respect to our results, but show the same overall trend, where the higher average Vp
values recorded in Maiolica, and the lower ones in the Tertiary marly Fms. This increase in P-wave velocity with depth is
primarily attributed to the increase in density and compaction of the rocks. Due to a different degree of porosity linked to the
different investigated depths and to the pore type, P-wave velocity can significantly change (e.g. Hairabian et al., 2014;
Smeraglia et al., 2014; Trippetta et al., 2021). Moreover, P-wave velocity also depends on factors such as lithology changes,
presence of fractures or faults and also on the different amount of tectonic deformation observed in different structural domains
(Trippetta et al., 2021).
Collecting and analysing the velocity data of upper crustal sedimentary rocks is very useful under different aspects. On one
end, these values help in building up and calibrating more accurate, 2D and 3D velocity models, that can be used for improving
the earthquakes localization (e.g. Latorre et al., 2016; Montone and Mariucci, 2023) as well as to constrain depth conversion
of seismic reflection profiles. On the other hand, since the velocity parameters of rocks are strictly related to their rigidity,
velocity values also reflect into their different mechanical behaviour and may ultimately influence earthquake generation and
distribution. In our case, the more competent carbonate formations (i.e. Maiolica and Scaglia Rossa Fms.) are characterised by
systematically higher velocity values with respect to the less competent, clay-rich turbidite formations (i.e. Schlier and
Marnoso-Arenacea Fms.).
In the ATF region, as well as in adjacent areas in the same seismotectonic framework of the Central Apennines, several recent
studies of the relationships between seismicity distribution and upper crustal geological setting have been recently performed,
by plotting accurately relocated seismic sequences on well-calibrated geological subsurface models, based on depth conversion
of seismic reflection profiles (e.g. Latorre et al., 2016; Barchi et al., 2021; Collettini et al., 2022; Chiaraluce et al., 2017b).
These studies coherently indicated that upper crustal seismicity (and the normal faulting earthquake mainshocks, in particular)
are systematically hosted in the high velocity, Mesozoic or Early Tertiary successions, consisting of carbonates, dolostones
and anhydrites, whilst only few, low magnitude events are recorded in the overlying, less competent Neogene turbidites, mainly
consisting of marls and sandstones.

**5.2 Stress Field**



As already mentioned, in the STAR study area (Fig. 2) two deep wells (SD and MC) were drilled in the past, reaching a depth
of 4763 and 5600 m, respectively. The SD well is located very close to the ATF, intersecting it at a shallow depth; while the
MC well, approximately 15 km east of the ATF fault, intersects additional tectonic structures. A thorough analysis of borehole
breakout stress data conducted along the two deep wells, allowed to deduce the current stress field orientation (Mariucci et al.,
2008). In the case of the SD well, the borehole breakout results reveal a minimum horizontal stress orientation of N055±22°;
the MC well exhibits a slightly different stress orientation, with a value of N012±29° (Fig. 11).





**Figure 11: Present-day stress field of Central Italy. The minimum horizontal stress orientations inferred both from the >45° dipping fractures detected along the STAR boreholes (bars with yellow triangles) and from the Italian Present-day Stress Indicators database IPSI 1.6 (Mariucci and Montone, 2024) are shown. 1- Stress data scaled by quality (from B to D) and coloured according to tectonic**



**regime: black bar is unknown regime (borehole breakout data), blue is thrust faulting, green is strike-slip faulting, red is normal faulting; 2- formal inversion data; 3- fault data; 4- data from TSM boreholes.**

To interpret the structural data obtained along the STAR boreholes, we have considered i) the shallow depth of the boreholes, ii) their near-vertical inclination, that strongly underestimates occurrence of near-vertical features (Terzaghi, 1965; Massiot et al., 2015), and iii) the difficulty in distinguishing fractures from faults in this type of data.

Along the STAR boreholes, the main features with a dip angle > 45° (Fig. 10, column f) show a predominantly NW-SE orientation and could be interpreted as pre-existing fractures generated in the early phases of the deformation history before folding (Price, 1966).

Considering that the current stress field in the area is primarily due to an extensional regime, with the main compressional stress, sigma 1, vertical and horizontal sigma 3 NE-SW oriented, the tectonic structures associated with this latter stress field consequently develop normal faults and high-dip extensional fractures. These > 45° NW-SE structures, favourably oriented with respect to the current extensional stress field (Fig. 11), might have been reactivated as extensional fractures. In TSM04 and TSM06 boreholes, for structures with a dip greater than 45° (Fig. 10, column f), another orientation is also observed, approximately 90° from the previous one. We can assume that both fractures NW-SE and NE-SW oriented are contemporaneous and linked to the same extensional stress field, primarily guided by a vertical sigma 1. On the other hand, structures with low dip - still NW-SE oriented - could be attributed to previous compressive deformation phases linked to a stress field characterised by a horizontal sigma 1, oriented NE-SW.

Taking into careful consideration the different depths investigated by the STAR boreholes (<0.2 km) with respect to the active stress data mainly inferred from breakout data in deep wells (0.5-6 km) and focal mechanisms of crustal earthquakes (usually 5-15 km), we can still compare their results (Fig. 11). In fact, most of the results from literature on the orientation of the stress field have shown that different crustal depths do not reorient or change the stress field (Heidbach et al., 2016; Mariucci and Montone, 2024). An almost constant orientation of the minimum horizontal stress characterises this sector of the Apennines, from the southern L'Aquila and Norcia zones to the areas north of Gubbio, showing only a slight rotation from ENE-NE directions to NE-NNE directions, respectively (Fig. 11). This is observed from numerous data derived from earthquake focal mechanisms as well as breakout data in deep wells, also present in the southernmost sector (Mariucci and Montone, 2016) and active fault data (Lavecchia et al., 2022), under a stress regime that can be defined as exclusively extensional.

In conclusion, our paper provides reliable values on physical properties of rocks, in particular P-wave velocity, which can be used to characterise crustal velocity models and allow detailed interpretation of seismic profiles, investigating the first two hundred metres of the crust.





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

## Data Availability

After a 3-year embargo period, starting from the end of drilling the last hole (TMS06 summer 2022), the
downhole logging dataset will be available at the ICDP repository database (https://www.icdp-
online.org/projects/by-continent/europe/star-italy/internal-data). Currently, the dataset is not available
for public download. Who is interested should contact the principal investigator of the ICDP STAR
project (Lauro Chiaraluce lauro.chiaraluce@ingv.it).

## Author Contributions

Author Contributions: PM, SP, MRB conceptualization, methodology; PM, SP, MTM, AA formal
analysis; PM, SP, MTM, LC, MU, FM, WJ field investigation; LC, AA, MU, FM data curation; PM,
SP, MRB, MTM writing—original draft preparation; SP, MU, MTM visualisation; PM, SP, MRB,



MTM writing—review and editing. All co-authors contributed to reviewing and revising the paper. All authors have read and agreed to the published version of the manuscript.

**Competing Interests**

The authors declare that they have no conflict of interests.

**Acknowledgments**

GEOTEC (http://geo-tec.it/en/) and GEOLOGIN Srl (https://www.geolog-in.com/) are thanked for providing drillings and geophysical borehole log data. We are grateful to Earthscope for providing strainmeters. STAR drilling project is co-funded by the International Continental Scientific Drilling Program (ICDP), by the United States National Science Foundation (NSF) and by the Italian Istituto Nazionale di Geofisica e Vulcanologia (INGV).

**Supplementary Material**

See Supplementary file