# Peer review of "Geophysical downhole logging analysis within the shallow depth ICDP"

_EGUsphere, 2024_

## Author Response (AR1)

Dear Editor,

We appreciate the referees' efforts and the valuable feedback they provided on our manuscript. The requested changes are all very relevant, and we agree with their suggestions. We will take them into account in the revised version of the manuscript.

RC1: Maria Beatrice Magnani 10 Jun 2024

The paper by Dr. Montone and colleagues presents new geophysical log data and analysis from six shallow (max depth 160 m) boreholes drilled as part of an ongoing project (STAR project) focused on monitoring the seismic and aseismic deformation on one of the most active low angle normal faults in the Northern Apennines, the Alto Tiberina Fault (ATF). The boreholes are located in the northwest portion of the actively extending region of the Umbria-Marche Apennines. This region of the Apennines has been the location of several, significant, deadly earthquake sequences, the last one of which began in 2016, culminating with a Mw 6.6.

The new data provide insights into the petrophysical properties and tectonic structure orientation at shallow depth of the lithostratigraphic units that make up the northern Apennines, from the late Mesozoic limestones of the Umbria-Marche pelagic units through the more recent siliciclastic turbiditic deposits. The Umbria-Marche carbonate multilayer is capable of hosting sustained levels of seismicity, and any direct information of the physical properties and fractures of these rocks is therefore relevant to understand their mechanical behavior. Additionally, these data are important because little is known about the lithologies at these shallow depths, and the petrophysical properties measured in situ can be then compared with other analogue measurements acquired in deeper boreholes or in lab experiments, allowing estimates of the effects of confining pressures and different stress conditions.The analysis of the orientation of the tectonic structures from the boreholes shows that they are favorably oriented for formation/reactivation in the present stress field, and the authors show that the stress field orientation at shallow depths persists unchanged throughout the crust in this sector of the Apennines.

The paper is well structured, written and illustrated, with an exhaustive explanation of downhole logging processing and analysis, and a clear description of results, and it is a relevant contribution to the Solid Earth community.

Minimal edits can improve the readability and clarity of the paper. Suggestions are listed briefly below.

I suggest that the physical properties of the different formations of the Umbria-Marche multilayer investigated by the boreholes be summarized in a table (for example by adding this information in Table 1).

> **We agree with the reviewer's suggestion and have included the name of the lithological formation in Table 1. Since most of the physical properties pertain to the open hole section of the boreholes, the lithologies refer only to this portion. We have therefore also specified this in the caption of Table 1: *The lithological formations refer to the open hole section. We have also added in the caption the meaning of n.d.: "n.d. stands for no data".**

In addition, it would be useful to know how the location of the boreholes was selected (which criteria guided the site selection and identification).

**We agree with the reviewer's suggestion and have modified the text accordingly by adding the following sentences to the Introduction and section 4.1:**

**Introduction:** *The locations of the six boreholes were selected based on two main criteria: on a regional scale, the objective was to enhance understanding of the seismic behaviour of the ATF; on a local scale, the primary requirement was to deploy the instruments while avoiding areas of inhomogeneous, anisotropic, or highly fractured rock volumes (Chiaraluce et al., 2024).*

**Section 4.1:** *Six STAR boreholes were drilled surrounding the creeping portion of the ATF to deploy strainmeters and seismometers (Fig. 1) in order to improve our understanding of the ATF seismicity pattern and monitor the evolution of seismicity in this area over time (Chiaraluce et al., 2024). For the location of our six boreholes we have followed two criteria: at a regional scale, three boreholes (TSM01, 02 and 06) are located following the axis of the Gubbio anticline and its northern continuation and the others (TSM03, 04 and 05) where the maximum extension rate is expected. At a local scale, the instruments were deployed in relatively homogeneous lithology, avoiding complex formations (e.g. alternating stiff and weak rocks or limestones and marl layers) as well as anisotropic or highly fractured rock intervals. The downhole logging measurements were performed immediately after drilling and prior to instrument installation. A posteriori, the known mechanical properties of the rocks hosting the instruments will be used to improve the interpretation of the data recorded.*
*The boreholes are located within an area of about 1500 km², centred around the town of Gubbio (Fig. 2 and Table S1). Among the six new boreholes, TSM01 and TSM02, near the Gubbio fault, were drilled across the late Mesozoic-Early Tertiary carbonates, cropping out along the crest of the Gubbio anticline, which represents the maximum structural culmination of the study area (Fig. 4 a, b). The other boreholes were drilled in the Tertiary marls and sandstones, cropping out in the northern part of the study area: TSM04 is located on the footwall of the ATF, TSM03 and TSM06 were drilled on the ATF hangiwall, and TSM05 is located in the farthest part of the ATF, close to the western flank of the Umbria-Marche ridge (Fig. 4 c, d, e and f;, for more detail, see Chiaraluce et al., 2024).*

Technical corrections:

Line 34 - EPOS initiative: Please spell out EPOS when using it for the first time.

**We agree and have added the meaning of the acronym EPOS (European Plate Observing System) to the text.**

Line 50 - "monitor and record THE seismicity of the low-angle...." (insert THE)

**We agree**

Line 52 - "measure small creep events": what is intended here by "small"? short duration? slow?

**To better explain the meaning of "small creep events", we added the following sentence:** *The small creep events can be recorded by strainmeters whose high resolution, for periods of hours to ~10 days, enables the identification of subtle, time-varying crustal deformations that are too small to be measured by GNSS or InSAR.*

Line 93 - replace "rock MASSES" with "rock VOLUMES"

**We agree and have changed accordingly**

Line 165 - replace "conductibility" with "conductivity"

**Thank you for pointing out this spelling mistake. We have changed it to *conductivity*.**

Line 205 - This section refers repeatedly to images (TT images, acoustic images, AMPL images) but there is no reference to a specific figure to better follow the text. I suggest the authors select a clear figure from one of the boreholes and use it for this purpose here.

**We agree. Since Figures 6 to 9, which display the TT and AMPL images, appear much later in the text, it seemed inappropriate to reference Figure 6 before Figure 4. To address this, we have revised it to: *see the figures related to the boreholes, e.g., Fig. 6.***

Line 275 - add "velocity" in "shear and compressional sonic logs VELOCITY to compare it to fracture porosity"

**We have added *wave velocity* and modified the sentence as follows: *shear and compressional wave velocity from sonic logs to compare..***

Line 368 - remove "accurately" and replace with "conduct ACCURATE geophysical downhole logs".

**We have modified accordingly**

Line 414 - remove IN and UP in "these values help IN building UP".

**We agree and have modified accordingly**

Line 417 - remove INTO in "reflect INTO their"

**We agree and have modified accordingly**

Line 419 - rephrase "higher velocitIES COMPARED to less competent".

**Thank you for the suggestion, we have modified it accordingly.**

Fig. 6 - there is some random text on the "Tadpole true" column. Remove

**We agree and have removed it.**

Fig. 9 - Missing rectangle highlighting the portion of the borehole shown on the bottom

**We agree and have inserted it.**

RC2: Anonymous Referee #2, 25 Jul 2024

The work presents a set of borehole data acquired as part of STAR, an ICDP project aimed at studying the seismic/aseismic behavior of the Alto Tiberina Fault, one of the most active faults in the northern Apennines, Central Italy. The wells house monitoring instruments that are part of the Near Fault Observatory TABOO, a multidisciplinary research infrastructure that is part of the European Plate Observing System. Well logging drive the positioning of the monitoring instruments in the boreholes.

In the absence of core sampling, well logs are the only means of deciphering the stratigraphy crosscut by the boreholes, and acquiring data on the physical properties of rocks formations. With this in mind, although the boreholes presented here are relatively shallow (max. 133 m depth), the new dataset is certainly relevant for completing the knowledge of the area under investigation.

The well-described and well-presented dataset is, however, a standard dataset, with no elements of technological innovation.

The authors state that the main objective of their work is to bridge the gap between the physical properties in the literature, obtained from rock samples, typically from outcrops, analyzed in laboratory, and geophysical field data or deep borehole data collected a few kilometres deep. Actually, it would be worth discussing differences and similarities of deep and surface boreholes in in more detail.

The data from deep and shallow wells discussed here, are comparable in terms of some parameters and not others. Why?

> **We were limited to comparing the data available to us. Since few parameters (e.g., GR, Vp and Resistivity) were available from deep wells, our discussion focussed more on these aspects.**

For example, the 30% difference between seismic velocity data measured in deep and shallow wells is not correlated with fracture density, but attributed to compaction. Are there data that can confirm that, e.g. a comparison with density increase? Can the authors quantify the role of fractures?

> **We have split our response into 3 parts:**
>
> **1 - We thank the reviewer for this valuable comment, which led us to re-examine the data and realise that the estimated percentage was incorrect. We then carefully rechecked the data from the literature and their statistical distribution (e.g., Trippetta et al., 2021). From this new analysis, the data from the STAR boreholes fall within the variability range of the deep wells (particularly similar to what is observed in the M. Civitello well, the only deep well available in the STAR area). Our values are approximately 15% lower compared to those of the deep wells; however, this difference is not particularly significant.**
>
> **Along the text we modified accordingly:**

*We compared the Vp values recorded in the six shallow STAR boreholes (depth < 0.2 km, see Table 1) with data derived from sonic log analyses in much deeper wells (depth > 4 km) drilled in the same region for industrial purposes (Bigi et al., 2011; Scisciani et al., 2014; Montone and Mariucci, 2020; Trippetta et al., 2021). These studies report average Vp values between 4.0 km/s and 4.8 km/s for the Marnoso-Arenacea Fm., 4.4 km/s to 4.8 km/s for the Schlier Fm., 5.3 km/s to 5.8 km/s for the Scaglia Rossa Fm., and 5.5 km/s to 6.1 km/s for the Maiolica Fm. All these values are approximately 15% higher than our results but follow the same overall trend, with higher average Vp values recorded in the Maiolica and lower values in the Tertiary marly Formations. This increase in P-wave velocity with depth is primarily attributed to increased rock density and compaction, particularly in the Tertiary formations.*

**2 - Regarding the second part of the reviewer's question: in the deep wells MC and SD, both derived and direct density data clearly indicate that density increases with depth. Figure 4 from the paper by Montone and Mariucci (Scientific Reports, 2020, 10:3834, https://doi.org/10.1038/s41598-020-60855-0) presents data for a series of deep wells in the Central Apennines. Specifically, the figure shows wells MC and SD labelled as 2 and 3, respectively. Density increases rapidly up to a depth of 3 km, reaching values of up to 2.8 g/cm³, and then stabilises and remains relatively constant.**

[Figure]

**3 - Regarding the last part of this question, we performed the Rock Quality Designation (RQD) analysis. The RQD is a procedure applied in the mining industry and is defined as a method to assess the quality and integrity of a rock mass by calculating the percentage of intact drill core pieces longer than 10 cm recovered during a single core run. We used the RQD tool in our software to analyse the image log to compute an RQD value based on the interpretation of ABI or OBI fractures data. The description of rock quality (RQD results) is displayed as values in a range from 0 to 1 (i.e. 0% to 100%). We have used the following ranking: 0.00 - 0.20 very poor quality; 0.20 - 0.40 poor quality; 0.40 - 0.60 fair quality; 0.60 - 0.80 good quality; 0.80 - 1.00 excellent quality.**

**Our results showed an RQD between 0.70 to 1.00. Based on this result, we supposed that the fractures likely did not significantly impact Vp values; rather, compaction could be a contributing factor, particularly for the Tertiary formations. Moreover, density log in the six shallow boreholes of STAR project has not been performed, therefore, we can only hypothesise and present this statement in a more tentative manner.**

The stress field inferred from breakout data in deep and shallow wells seems to remain the same. This is an important result. How widespread is this observation, and how does it differ from other cases where the stress field varies as one approaches the surface?

**We apologise and we may not have been clear in the text, but we would like to specify that the available breakout data refers only to results obtained in the past from the deep wells SD and MC. In the STAR project, the boreholes reach a maximum depth of 160 m, and from the image analyses (OBI and ABI logs) no breakouts were detected in any of the boreholes. If there were, they could not be related or representative to the stress field at depth because their occurrence would be linked to local effects such as topography.**

**As discussed in numerous studies (e.g. Zoback et al., 1989; Zoback 1992; Pierdominici and Heidbach, 2012), the orientation of the crustal stress field, in areas characterised by the same tectonics, remains constant both with depth and spatially. Near the surface, however, the stress field may be influenced by local faulting, folding, and other near-surface geological structures, as well as by topographical features and erosion, which can cause its orientation to vary. This is one of the reasons why breakout analysis is usually not performed in the first 500 metres of depth, as its orientation can differ from the stress regime at depth. Rotations of the stress field orientation have been observed along some deep wells, particularly near active tectonic structures (e.g. Bell et al., 1992; Mariucci et al., 2002; Pierdominici et al. 2011; Pierdominici et al., 2020). In these cases, the integration of multiple datasets can provide a more comprehensive understanding of the stress field at different depths. Based on these considerations and the total data available on the present-day stress field, we can constrain the orientation of the stress field in this sector of the Apennines to a predominantly extensional kinematics with maximum extension oriented in the NE-SW direction, both at depth and spatially.**

**To be clearer in the text, we have included a sentence (section 5.2):**

***To constrain the orientation of the stress field in the area of the STAR project, we have analysed and interpreted the fractures detected in the six boreholes and compared them with the breakout orientations from the two deep boreholes and also with the other stress indicators, mainly focal mechanism data. In our boreholes, ABI and OBI images do not reveal any borehole breakouts. However, breakouts at very shallow depth could not be related or representative to the stress field because their occurrence would be linked to local effects such as topography, local faulting, folding, and other near-surface geological structures.***

My overall impression is that as it stands, the manuscript presents a valuable collection of data, which reconfirms the knowledge already established in the literature but is a bit lacking in scientific novelty. As the manuscript is at present it would perhaps be more suitable for a data journal and not for a research journal. I would recommend a complete revision of the discussion of the results that highlight the novelty brought by this dataset, for example highlighting how the

data drives the build up of a near fault observatory (e.g. how were the boreholes locations selected?), or how are the implications in the view of seismic/aseismic behavior, or also how do they close the gap between lab and field data e.g. discuss the scale difference, the upscaling problem, the different constrains.

**Thank you for your recommendation. We have revised the text and in particular the discussion section to highlight the novelty brought by this dataset. Specifically, we have explained the selection of borehole locations detailing how the borehole locations were chosen to build a near-fault observatory, including the criteria and rationale behind their selection. Moreover, we discussed how our dataset bridges the gap between laboratory, field data, deep and shallow geophysical measurements.**

**We hope these revisions aim to clearly demonstrate the significance of our dataset within the proposed scientific approach and its impact on advancing knowledge in the field.**

**In detail:**

**Regarding the "boreholes locations selected", we added along the Introduction the following sentence (as also suggested by the other referee)*:***
*The locations of the six boreholes were selected based on two main criteria: on a regional scale, the objective was to enhance understanding of the seismic behaviour of the ATF; on a local scale, the primary requirement was to deploy the instruments while avoiding areas of inhomogeneous, anisotropic, or highly fractured rock volumes (Chiaraluce et al., 2024).*

**Moreover, we added in the Section 4.1 the following sentences:**
*Six STAR boreholes were drilled surrounding the creeping portion of the ATF to deploy strainmeters and seismometers (Fig. 1) in order to improve our understanding of the ATF seismicity pattern and monitor the evolution of seismicity in this area over time (Chiaraluce et al., 2024). For the location of our six boreholes we have followed two criteria: at a regional scale, three boreholes (TSM01, 02 and 06) are located following the axis of the Gubbio anticline and its northern continuation and the others (TSM03, 04 and 05) where the maximum extension rate is expected. At a local scale, the instruments were deployed in relatively homogeneous lithology, avoiding complex formations (e.g. alternating stiff and weak rocks or limestones and marl layers) as well as anisotropic or highly fractured rock intervals. The downhole logging measurements were performed immediately after drilling and prior to instrument installation. A posteriori, the known mechanical properties of the rocks hosting the instruments will be used to improve the interpretation of the data recorded. The boreholes are located within an area of about 1500 $km^2$, centred around the town of Gubbio (Fig. 2 and Table S1). Among the six new boreholes, TSM01 and TSM02, near the Gubbio fault, were drilled across the late Mesozoic-Early Tertiary carbonates, cropping out along the crest of the Gubbio anticline, which represents the maximum structural culmination of the study area (Fig. 4 a, b). The other boreholes were drilled in the Tertiary marls and sandstones, cropping out in the northern part of the study area: TSM04 is located on the footwall of the ATF, TSM03 and TSM06 were drilled on the ATF hangiwall, and TSM05 is located in the farthest part of the ATF, close to the western flank of the Umbria-Marche ridge (Fig. 4 c, d, e and f;, for more detail, see Chiaraluce et al., 2024).*

**In the Introduction we added:**

*Beyond the specific case reported here, geophysical and petrophysical data acquired in shallow boreholes also contribute to the knowledge of the upper crust. Indeed, they provide in situ measurements linking those obtained on outcropping rocks, often influenced by surface processes, with those from deep wells. Our study shows how even with relatively few data in a large area, valuable insights can be obtained. With more data in a region of interest, we would be able to shed a light on a complete portion of the crust, from surface to few kilometres depth, or even deeper.*

**At the end of Paragraph 5:** *Most geophysical and petrophysical data available in the literature come either from rock samples, analysed in the laboratories, or from well-logs, acquired along deep wells, whereas our data from geophysical logging of shallow boreholes provides an almost untapped source of information.*

**At the end of paragraph 5.2:**

*However, detailed geophysical measurements from shallow boreholes are relatively rare. A small, homogeneous rock sample analysed in the laboratory may not accurately represent the complexity of the in situ rock formation, which can exhibit significant internal variability in composition and fracturing. Additionally, although our sites provide in situ measurements, they represent a relatively small dataset compared to the extensive data collected from deep well logs, which span hundreds of metres. This limitation could explain the discrepancies observed, such as the differences in Vp values. While our measured values are lower than the average data, they still fall within the acceptable range.*
*Beyond the specific case presented, our data significantly enhance our understanding of the upper crust. These in situ measurements bridge the gap between data from outcropping rocks and data from deeper wells. In particular, this scientific approach is able to provide useful geophysical information at the very shallow crustal depth (<0.2 km), typically not explored by either the scientific community or oil and gas industry. Our study demonstrates that even a limited dataset can provide valuable insights and a basis for future projects. With an expanded dataset across a region of interest, it would be possible to illuminate a comprehensive section of the crust, extending from the surface to several kilometres deep, and potentially even deeper.*

**To make an example, we can consider figure 4 from Trippetta et al., 2021, and showing the Vp measurement in the Maiolica fm.**

[Figure]

Figure 4. (A) In situ Vp sonic logs (SLV) depth profiles of boreholes through Maiolica (MA). (B) Velocity frequency cumulative histograms of the boreholes in panel A; y-Axis shows the number of times (frequency) that a certain velocity was recorded. Inset shows the frequency analysis of a representative single well (SpinelloMare1). (C) Interval velocities (IV) depth profile for available boreholes for the MA portions where vertical bars indicate the drilled thickness. Note that IV instrumental error is in the order of 2% (based on estimates in previous literature)[47] and that error bars are reported in Fig. 6.

**In the Trippetta diagram it is possible to see that no measurement is available at depth < 1.5 km, then our results fill this gap representing a novelty of our study. Moreover, the value recorded at TSM01 (5.3 km/s +-0.4 = 4.9 km/s to 5.7 km/s) is fully consistent with the measurements in deep boreholes. The average value (5.3 km/s) is less than 10% lower than the average measured values. Similar observations can be done for Scaglia Fm. (4.9 km/s +- 0.2 = 4.7-5.1 km/s).**

**These observations point to the conclusion that, in the shallow crust, once the diagenesis process of carbonate rocks is complete, depth (pressure) cannot induce further significant compaction or a substantial increase in wave velocity values, unless metamorphic processes are initiated at greater depths. Significantly lower Vp values were observed in the Marnoso-Arenacea Fm. Here, we drilled mostly marly successions, trying to avoid stronger sandstones levels, selecting the softest lithology. Moreover, the clayey sediments may have been de-compacted during recent exhumation of these rock formations.**

Punctual observations:

Fig. 2: it would be nice to have the ATF located.

**We agree and we have added the ATF in figure 2.**

Fig. 4: Legend: the Marnoso-Arenacea formation is reported with 5 different colors. Why? In Fig 5 the Marnoso-Arenacea is only one. which one is it?

**The Marnoso-Arenacea Fm. encompasses several geological members, which are indicated with different colours and patterns as shown in Figure 4.**
**In Figure 5, the Marnoso-Arenacea is depicted as a single formation without subdivision into members. We did not find it necessary to differentiate between various members in**

**Figure 5 for two reasons: 1. The reader can identify the specific members associated with each borehole based on Figure 4 (TSM03 corresponds to member FMA3; TSM4 to member MUM; and TSM6 to member FMA4); 2. The gamma-ray (GR) measurements do not show significant variations depending on the specific member within the Marnoso-Arenacea Formation**

Fig. 5: Unit of the GR missing.

**Thank you for pointing out this omission. We have added the unit [cps] in the figure and specified its meaning (cps stands for counts per second) in the caption.**

Fig. 6: in the column Tadpole true there is a text that should be cancelled (Lorem ...)

**Thank you for pointing that out!**

Fig. 8, 9: what are the N?

**Thank you for this comment. We have added the meaning of N in the caption of fig. 8 and Fig. 9. N means normal resistivity. The numbers [8, 16, 32 and 64] indicate the distance in inches between the electrode reference measuring point to the injection electrode. Explanation has been added in the "Supplementary 2: Logging Tools description".**

L 82: "Giving food for thoughts about the effects" sounds a colloquialism. Better perhaps "that give insights into the effects"

**We agree and we have followed reviewer´ suggestion and have re-phrased as *that give insights into the effects*.**

L 546: The title of De Paola et al. 2009 (https://doi.org/10.1029/2008JB005967) is "Brittle versus ductile deformation as the main control on the transport properties of low-porosity anhydrite rocks"

**We apologise and have changed the title accordingly**

Citation: https://doi.org/10.5194/egusphere-2024-1249-RC2

CC1: Giacomo Medici 19 Jul 2024

General comments

Good and novel research in the field of structural geology/geophysics. I am providing minor comments as researcher working on download borehole logging and petrophysics.

Specific comments

Lines 11-26. Make clear the depth of investigation for the boreholes studied in the abstract. Researchers by seeing wireline and fluid logs might get the impression of a much deeper investigation.

> **We agree, we have added:** *maximum depth 160 m*

Lines 30-97. I would push the introduction beyond literature on STAR drilling project and the regional geology. I'm only trying to bring the impact out of your novel research.

> **Thank you for the comment. We have revised a bit the Introduction adding a few sentences:**
>
> *Beyond the specific case reported here, geophysical and petrophysical data acquired in shallow boreholes also contribute to the knowledge of the upper crust. Indeed, they provide in situ measurements linking those obtained on outcropping rocks, often influenced by surface processes, with those from deep wells. Our study shows how even with relatively few data in a large area, valuable insights can be obtained. With more data in a region of interest, we would be able to shed a light on a complete portion of the crust, from surface to few kilometres depth, or even deeper.*

Line 97. Consider to disclose the 3 to 4 specific objectives of your research by using numbers (e.g., i, ii, and iii).

> **Regarding this comment, the specific objectives are referred to the STAR project and not to this study "the outcomes from the STAR drilling project will provide a better understanding of the behaviour of the main active faults, addressing fundamental questions about the relationship between creep, slow slip, dynamic earthquake rupture, and tectonic faulting". We believe that using i., ii., iii. to emphasize the general outcomes of the STAR project, rather than focusing on our study, could be confusing for the reader.**

Lines 156-157. "Data acquisition from downhole logging becomes the key element in determining the rocks physical characterization". Add recent review that incorporates techniques for downhole logging methods in fractured rocks:

- Review of Discrete Fracture Network Characterization for Geothermal Energy Extraction. Frontiers in Earth Sciences 11, 1328397

> **Thank you for suggesting this review paper. We have added the article.**

Line 292. Can you provide information on the speed of downhole logging for OBI and ABI? It can impact of the quality of the dataset and the structure picking analysis. Other data available have much lower resolution in the same geological formations. Researchers might be interested on finding the reason.

**Thank you for this comment. The speed of the OBI and ABI in the STAR project was approximately 2 m/min. This speed is the one normally adopted in scientific drilling projects because it allows a good image to be obtained. However, clarity of the image log is not only related to the speed but there are other elements to be taken into account such as the transparency of the drilling fluid. In STAR project we operated in muddy formation (especially for the Schlier and Marnoso-Arenacea Fms), so the boreholes were flushed several times until the return of the drilling fluid was clear so that we were sure that the borehole walls were clean from smearing mud during the drilling operations.**
**In the "Supplementary 2: Logging Tools description" we have added the following sentence:** *The speed of ABI is generally of 2 m/min. The clarity of the image logs, however, is not only related to the speed but there are other parameters to be taken into account such as the transparency of the drilling fluid***.**

Lines 368-470. Same here. I would extend the introduction beyond literature on STAR drilling project and the regional geology. I'm only trying to bring the impact out of your novel work.

**We agree, we have added some sentences about it also in the Conclusions.**

*However, detailed geophysical measurements from shallow boreholes are relatively rare. A small, homogeneous rock sample analysed in the laboratory may not accurately represent the complexity of the in situ rock formation, which can exhibit significant internal variability in composition and fracturing. Additionally, although our sites provide in situ measurements, they represent a relatively small dataset compared to the extensive data collected from deep well logs, which span hundreds of metres. This limitation could explain the discrepancies observed, such as the differences in Vp values. While our measured values are lower than the average data, they still fall within the acceptable range.*
*Beyond the specific case presented, our data significantly enhance our understanding of the upper crust. These in situ measurements bridge the gap between data from outcropping rocks and data from deeper wells. In particular, this scientific approach is able to provide useful geophysical information at the very shallow crustal depth (<0.2 km), typically not explored by either the scientific community or oil and gas industry. Our study demonstrates that even a limited dataset can provide valuable insights and a basis for future projects. With an expanded dataset across a region of interest, it would be possible to illuminate a comprehensive section of the crust, extending from the surface to several kilometres deep, and potentially even deeper.*

Lines 472-652. Integrate recent and relevant scientific literature on the topic.

**We have added the following references:**

*Vuan, A., Brondi, P., Sugan, M., Chiaraluce, L., Di Stefano, R., and Michele, M.: Intermittent slip along the Alto Tiberina low angle normal fault in central Italy, Geophys. Res. Lett., 47, e2020GL08903, https://doi.org/10.1029/2020GL089039, 2020.*

*Anderlini, L., Serpelloni, E., and Belardinelli, M.: Creep and locking of a low-angle normal fault: Insights from the Altotiberina fault in the Northern Apennines (Italy), Geophys. Res. Lett., 43, 4321–4329, https://doi.org/10.1002/2016GL068604, 2016.*

*Mcginnis, R. N., Ferrill, D. A., Morris, A. P., Smart, K. J. & Lehrmann, D. Mechanical stratigraphic controls on natural fracture spacing and penetration. J. Struct. Geol. 95, 160–170 (2017).*

**We have aimed to select references that are closely related to the area and the topic discussed. Unfortunately, it is not easy to find relevant literature that contains information or data on carbonate rocks drilled in boreholes. Many studies refer to metamorphic and or geothermal environments and are not relevant for our manuscript. Additionally, we have included a few more references (Vuan et al., 2020; Anderlini et al., 2016; Mcginnis et al., 2017).**

**The cited 74 works, some of which (prior to the 2000s) represent milestones relating to the geology of the area, but also to the various topics covered in this study that cannot be left unmentioned. One third of the total number of citations relate to the period 2000-2019, where new technologies, interactions and data were studied, and one third from 2000 to the present day, which represent the most recent updates strictly related to our subject. We are aware of recently published papers on fractures, interpretation of downhole logging, geology of the area, but we thought to select and cite only those strictly relevant to our study, among the recent papers cited in our manuscript e.g.: Massiot et al, 2015, one of the novelties of this study is the new classification of fractures, that we have taken into account; Mariucci and Montone, 2024 for the update on the stress field in Italy; Pierdominici and Kück´s paper is an updating on downhole logging from an instrumental and interpretation; Montone and Mariucci, 2023 is a review on Vp velocity in deep wells in Italy; Barchi et al, 2021 is related to the geology of central Italy; as well as the papers of Collettini et al., 2022 and Trippetta et al. 2021 on the physical properties of rocks.**

Figures and tables

Figure 5. Variability of the thickness / lateral extension of the lithological logs is unclear.

**We have added the cps unit for GR in fig. 5 to clarify the lithological log.**

**In detail, Figure 5 (e.g., TSM1 borehole) shows a lithological sketch indicating different lithologies at depth. In the middle, we have included drilling information, such as the diameter of the hole and markings for cased and open sections. On the right side, the GR curve is displayed, reflecting the encountered lithology. In the Maiolica Formation, where limestone is the primary component, the GR values are very low (around 8 cps, shown in blue; see also Table 1). Different shades or peaks in the curve can indicate thin clay layers. High GR values (approximately 65 cps) are recorded from around 70 m to**

**about 58 m in the Marne a Fucoidi, which is rich in clay components and clearly marks the boundary between limestone and clay formations. As we move upwards, the clay component decreases, transitioning into the Scaglia Bianca (with calcareous components). The boreholes in the Marnoso-Arenacea Formation (TSM3, TSM4, and TSM6) show high GR values in the open hole (predominantly yellow with some red intervals) and lower values in the cased hole (blue). The blue colours in the cased hole do not indicate a reduced clay component but rather result from the steel casing interfering with the GR signal, thus producing lower values. Without the casing, the Marnoso-Arenacea Formation would exhibit the same high GR values as seen in the open sections. The GR values are related to the type of clay, sandstone, or limestone component, not to the thickness or lateral variability**

Citation: https://doi.org/10.5194/egusphere-2024-1249-CC1